# Manufacturing Methods Induced Property Variations in Ti6Al4V Using High-Speed Machining and Additive Manufacturing (AM)

Shivam Pradeep Yadav [1,*] and Raju S. Pawade [2]

1 Mechanical Engineering, Clemson University, Clemson, SC 29634, USA
2 Department of Mechanical Engineering, Dr. Babasaheb Ambedkar Technological University, Lonere 402103, Maharashtra, India
* Correspondence: shivamy@clemson.edu; Tel.: +1-(864)-7940645

**Abstract:** Additive manufacturing techniques are replacing conventional subtractive machining processes; however, the surface quality and defects have been a key roadblock to expanding AM's uses. This paper describes experimental investigations in the high-speed dry machining and additive manufacturing (AM) of titanium alloy (Ti6Al4V), discussing the effect of machining and AM conditions on the surface characteristics due to the micro-deformation layer. Analysis of the machined surfaces shows the deposition of microparticles at a high cutting speed of 170 m/min at moderate feed rates. The predominant thermal softening effect at a high cutting speed causes restructuring of the micro-deformation layer. Thus, the machined surface shows fewer alterations and a correspondingly lower surface roughness. A high cutting speed also favors the induction of high residual stresses that are compressive. Shallow grooves are seen throughout the surface along the feed spacing with a higher depth of cut of 0.8 mm. An increase in the cutting speed from 170 m/min to 190 m/min leads to a 61% increase in the surface finish owing to a rise in machining temperature leading to thermal softening, and subsequent restructuring of the machined surface layer occurs. For the feed rate, the surface finish values decrease gradually as the feed rate increases, and the worst finish of 1.37 μm is attained at a feed rate of 0.875 mm/rev. This study also compares different AM processes for Ti6Al4V based on the defects and their effects on mechanical properties, such as tensile and fatigue strength. It was observed that the ultimate tensile strength and the yield strength were approximately 20% more in SLM and direct energy deposition as compared to electron beam melting. The mechanism of these effects is also explained by elaborating on the influence of grain size, phase, and other microstructural behaviors on the final mechanical properties of the produced part.

**Keywords:** Ti6Al4V; high-speed machining (HSM); selective laser melting; electron beam melting; direct energy deposition

## 1. Introduction

The definition of high-speed machining (HSM) is complex. It differs from material to material, which is, the range of the machining speed in high-speed machining for aluminum begins above 500–1000 m/min, whereas it is 10–50 m/min for superalloys. The behavior of work material changes with the change in cutting speed; therefore, the transition of chips from continuous to shear localized is also different for different materials. However, the magnitude of the cutting forces and the temperature declines once the transition range is over. It is known that the use of flood coolant supplied at the chip–tool interface is less effective at a higher cutting speed. This is because of the higher magnitude of the centrifugal forces, which causes difficulty in reaching the coolant droplets in the small gap of the tool–chip interface. Further, the stringent regulations led by the government on the use of coolant in machining also compel manufacturers to carry out machining without coolant [1–8].

*1.1. Analysis of Ti6Al4V in High-Speed Machining*

To date, researchers have attempted to improve titanium alloy's machinability, especially Ti6Al4V, in the turning process. There are attempts reported in the literature focused on improving titanium's machinability [9–19]. It has been found that the machining difficulties are partially relieved at high cutting speeds owing to a reduction in cutting forces, a better surface finish, and surface damage. The use of cutting fluid demonstrates a more significant role in the dissipation of heat from the machining zone to the surroundings than any other means. However, cutting fluid's application with flood coolant has several disadvantages. In flood cooling applications, many resources are required, causing more energy consumption. Further, the application of cutting fluid represents 16–20% of the manufacturing cost, problems of procurement, storage disposal, and maintenance, grave contamination, and damage to the environment in all stages from confectionary to applicatory. For example, in the case of oil mist, smog mist ambiance and chemical particles in the shop floor led to more diseases and even cancer [16,19]. However, dry machining eliminates almost all the problems associated with flood coolant application during machining. A machinability study of Ti6Al4V was conducted to obtain a higher surface integrity with a suitable combination of the depth of cut, feed rate, and cutting speed using dry machining. However, very little knowledge is available on the use of dry machining for this material. These studies are insufficient to understand the surface generation mechanism and the consequent surface integrity when coolant is absent [9,12,14,20,21]. It is observed that the machining mechanism strongly impacts the temperature increase in the machining zone due to the absence of coolant. A microscopic investigation into the dry machining of titanium alloy is currently available in the literature. Given this, research has examined the benefits of dry machining over flood cooling during machining. Aluminum tends to stabilize the alpha phase and helps to increase the beta-transus temperature. Some of the reasons for the poor machinability of titanium alloys are a low elastic modulus, a short contact length between the chip and the tool, and the generation of the high cutting temperature of these alloys. Another problem encountered is the small heat-affected zone acting closer to the cutting edge while machining due to thinner chips produced because of the short contact length between the chip and the tool in addition to the presence of a fragile flow zone between the tool and the chip. Further, the low modulus of elasticity of these alloys proves to cause chatter formation during finish machining. Saraf and Shivam [21,22] studied the application of titanium alloys to manufacture stents. The investigations included the strength of manufactured stents made up of other materials and correlating it with titanium alloys. The factors of biodegradation, radial strength, elastic radial recoil, flexibility, trackability, stent profile, and scaffolding effects were studied for titanium. After machining these titanium alloy stents, the consequences led to grain size and orientation changes along with the crystal structure and microstructure. The discussion of titanium alloys for incorporating biomedical applications of bovine medial collateral ligaments is also seen. Zoya et al., 2000 [14], carried out turning experiments on a high-speed, precision lathe on an $\alpha$ + b phase-stabilized titanium alloy, the stabilizers being Al (4.5%) and Mn (4.5%) with CBN tools [23]. Above a 185 m/min cutting speed, the surface roughness increased exponentially. Ezugwu et al., 2005 [12,24], during experimental studies on titanium machining with different CBN tools, found that the surface finish decreases with an increased cutting speed. Pawade et al. [19,25] studied the high-speed turning of titanium alloy while optimizing the machining parameters and quantified the machining performance based on the damage, surface roughness, and chip thickness ratio. Significant changes in the surface finish were observed while varying the feed rates as compared to fluctuating cutting speeds. Ribeiro et al. (2003) [26] evaluated the effect of the cutting speed on the surface finish of titanium alloy during machining with conventional uncoated carbides. They observed an increase in the surface finish as the cutting speed increased from 55 m/min to 90 m/min, but the finish drastically deteriorated at 110 m/min. When machining was continued to a 160 m length, the surface roughness was more minor

until a machining length of 80 m, after which the cutting tool wear increased, deteriorating the finish of the Ti6Al4V alloy.

*1.2. Why Additive Manufacturing (AM) of Ti6Al4V Alloy*

Manufacturing titanium alloys with conventional manufacturing techniques requires higher tooling costs because of the direct interaction between the material and the manufacturing tools; hence, the cost increases significantly. Most cutting tools are made of titanium or ceramics in conventional machining, leaving us with inferior options to machine titanium alloys. Most of the tools are titanium coated. In addition, it is not advisable to use the same cutting tool material as the base material due to diffusion and seizure of their relative motion. Due to its high strength, making an excellent titanium structure is unnecessary. A bionic topologically optimized design can easily replace a solid design made from a different material, saving a lot of material and making the component super light; however, this feature is also not possible in conventional manufacturing methods, and hence, AM is preferred. In addition, titanium's density is less than steel, tungsten alloys, and copper alloys with a high strength, which makes it an ideal candidate for aerospace and biomedical applications.

Though AM is an expensive technique as compared to conventional manufacturing, in the case of manufacturing titanium, AM is more suitable because it can be manufactured near to the desired geometry in one pass, and the material waste is reduced (low buy-to-fly ratio) as well as the cost of using multiple tools, and handling charges are also minimized. Titanium's low density and high strength properties are demanded in all engineering and biomedical applications, from bridges to cardiovascular stents. However, there are certain challenges pertaining to printing Ti6Al4V:

- Poor thermal conductivity leads to higher porosity in AM;
- Strain hardening proclivity;
- Susceptibility to react with oxygen actively.

The thermal features lead to residual stress, which hampers the mechanical and fatigue properties of the end product. The typical thermal features affecting the microstructure of Ti6Al4V are:

1. High heat input localization;
2. Minimum time of interaction;
3. High-temperature difference between layers;
4. Superior cooling rates.

There is a clear gap in the literature that no such study has conducted previously where the properties of Ti6Al4V are compared to manufacturing it using conventional and additive manufacturing techniques. Such a study is instrumental in industries producing additively manufactured parts, employing machining as a post-processing technique to convert their product into a finished good, especially in orthopedic device manufacturing and aerospace component manufacturing. Therefore, it is necessary to precisely understand the correlation of the process parameters and their effects on the properties and microstructure of the manufactured part so that the following process can have an enhanced compatibility and can eliminate the risk of deteriorating the surface or mechanical properties of the component. Therefore, understanding the effects of the vital process parameters, such as laser power and scan speed variations, in different additive manufacturing processes plays a role in the microstructures, such as phase and grains, due to different cooling rates and temperature gradients and correspondingly in studying the effects of this microstructural variation on the surface roughness, hardness, and residual stresses induced in the manufactured components. Comparing the resulting variation in hardness and surface roughness with highspeed machining and analyzing the microstructural deviations that lead to such aberrations will help to understand and improve the compatibility between the two processes. Therefore, this article presents a comprehensive study to compare the tensile and fatigue properties of Ti6Al4V manufactured by various AM processes, such as

selective laser melting (SLM), electron beam melting (EBM), and direct energy deposition (DED), and to analyze its compatibility and property variations compared to conventionally manufactured Ti6Al4V and find a relation for these properties with the microstructures formed during these processes for load-bearing structures.

## 2. Materials and Methods

There are many different designs to fit response surfaces, viz., central composite design (CCD), Box–Behnken designs, etc. The sample of the Ti-6Al-4V alloy was received in the form of a rolled cylindrical rod. In our experiments, the rod diameter used for the turning operation was $\phi$16 mm and 80 mm long using a physical vapor deposition (PVD)-coated TiAlN-coated tungsten carbide cutting tool with dry conditions at high cutting speeds. Our study also correlates the machining parameters with the surface finish and its effects on the other factors discussed above. After machining, the surface topography and the surface and subsurface integrity were assessed using the measurement of residual stresses induced, surface finish, microhardness, and other surface alterations in the machined surfaces. The cutting tool geometry is shown in Figure 1b. The insert has a specification CNMG 120408 MP with KC5010 grade (Kennametal Widia, Pittsburg, PA, USA). KC5010 grade is especially suitable for heat-resistant materials, such as superalloys and titanium alloys. The machine used for the turning experiments is shown in Figure 1a, which is a Micromatic-made CNC Lathe (Model JOBBER XL$_1$, ACE Designers, Bangalore, Karnataka, India). The experiments were carried out according to the CCD array of RSM, where the machining process parameters included the depth of cut, feed rate, and cutting speed; each was varied at five levels. After machining, the surface topography and the surface and subsurface integrity were assessed using measures of the surface roughness, surface alterations, induced residual stresses, and microhardness of the machined surfaces. To determine the degree of work hardening, microhardness testing was carried out using the Shimadzu microhardness tester Make-Shimatzu (Model MV2) equipped with a precision X-Y table shown in Figure 1d. The indentations were performed with a Vickers indenter. The total included angle on the indenter tip was 136 degrees.

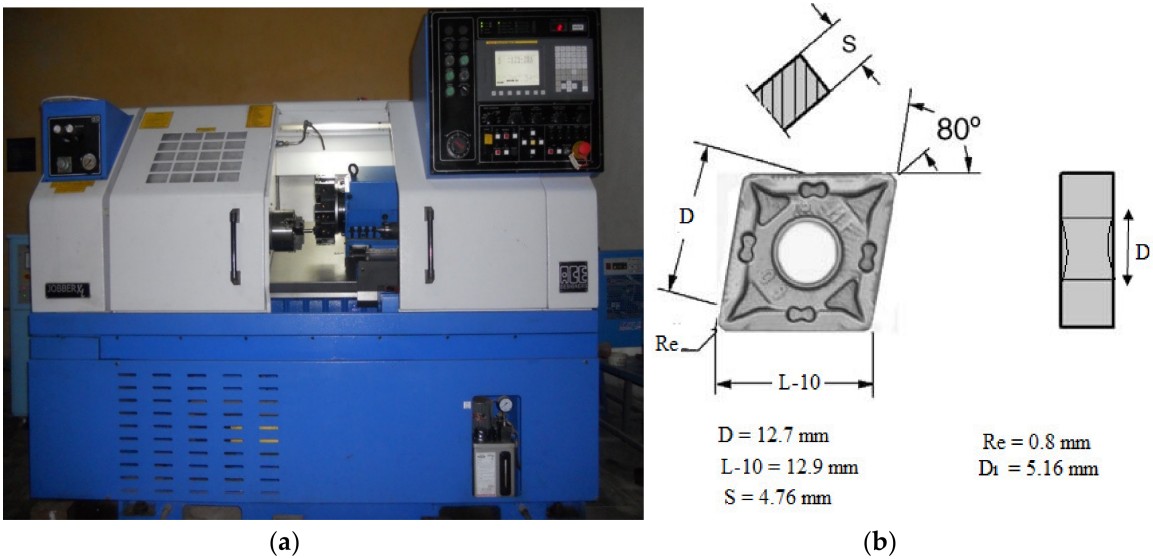

(**a**)  (**b**)

**Figure 1.** *Cont.*

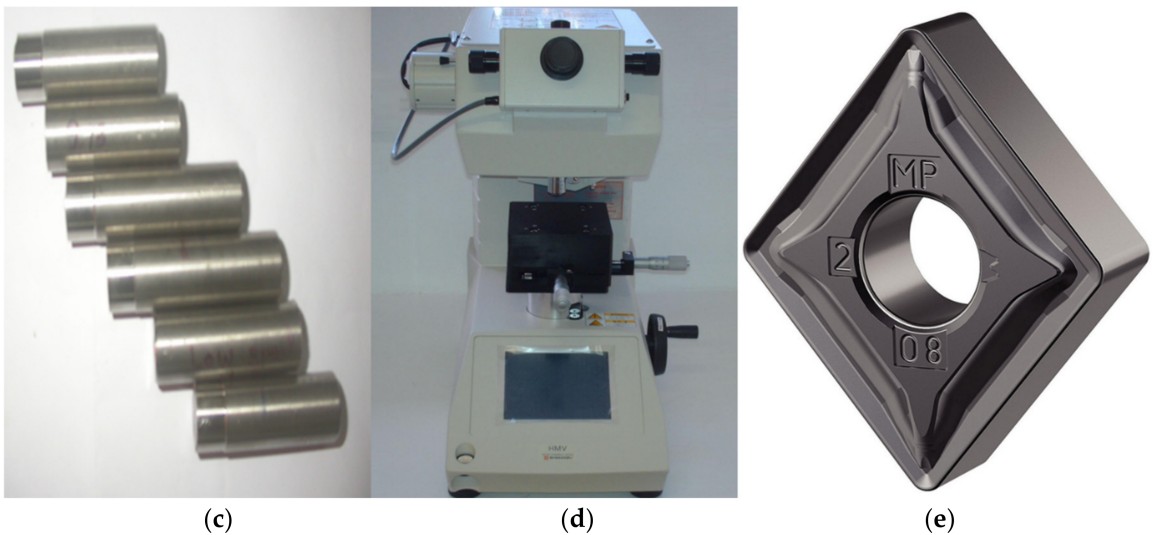

(c)            (d)            (e)

**Figure 1.** (**a**) CNC lathe used for experimentation; (**b**) geometry of the CNMG 120408 MP inserts; (**c**) Ti6Al4V work pieces; (**d**) Shimatzu microhardness tester; and (**e**) CNMG 120408 MP cutting insert.

Popular powder-based methods typically used for Ti6Al4V were selected to create a consistent comparison. The features of various AM processes are listed in Table 1.

**Table 1.** Features of AM processes used to manufacture Ti6Al4V.

| Selective Laser Melting | Direct Energy Deposition | Electron Beam Melting |
|---|---|---|
| 1. Moving build platform<br>2. Small layer thickness compared to DED<br>3. Unused powder cannot be used without recycling due to altered physical properties | 1. Moving laser and power feeding head<br>2. Controlled atmosphere with argon gas and oxygen levels below 5–10 ppm<br>3. Freedom to vary the composition during the process is possible<br>4. Less waste of material compared to SLM and EBM | 1. Electron beam used for the process<br>2. Vacuum environment of $10^{-4}$ bar pressure<br>3. High build Temperature of 600–750 °C<br>4. Fast build rate compared to SLM and DED<br>5. Beneficial method to manufacture reactive metals<br>6. Low surface finish |

## 3. Results and Discussions

### 3.1. Discussion of Surface Alterations and Hardness in High-Speed Machining

The micrographs show that the smeared layer was formed on the machined surface at high cutting speed conditions. As experiments were conducted at high cutting speeds, i.e., ranging from 150–190 m/min, these alterations were apparent because of the elevated temperature generated in the machining zone that caused the predominant plastic deformation due to the thermal action of the near-surface layers, see Figure 2c,d,f,h. The smeared layer was more severe in the case of experiments conducted at a higher depth of cut of 0.7, as can be seen in Figure 2f,h.

Microparticle deposits are tiny chip fragments because of severe friction between the underside of the chip surface and the tool rake face. They may be produced due to the rupture of the work-hardened surface layer of the machined surface when it comes in contact with the cutting tool. It is evident from the SEM analysis that microparticle deposits are noticeable at higher feed rates and cutting speeds. Microparticles are visible at a cutting speed of 170 m/min in Figure 2c,e, which occur as a result of the increment in the cutting speed that leads to the fragments of the chip material formed being ruptured.

In the deformation zone, the strain and strain rate increase simultaneously, increasing the plastic deformation of the chips generated from the workpiece.

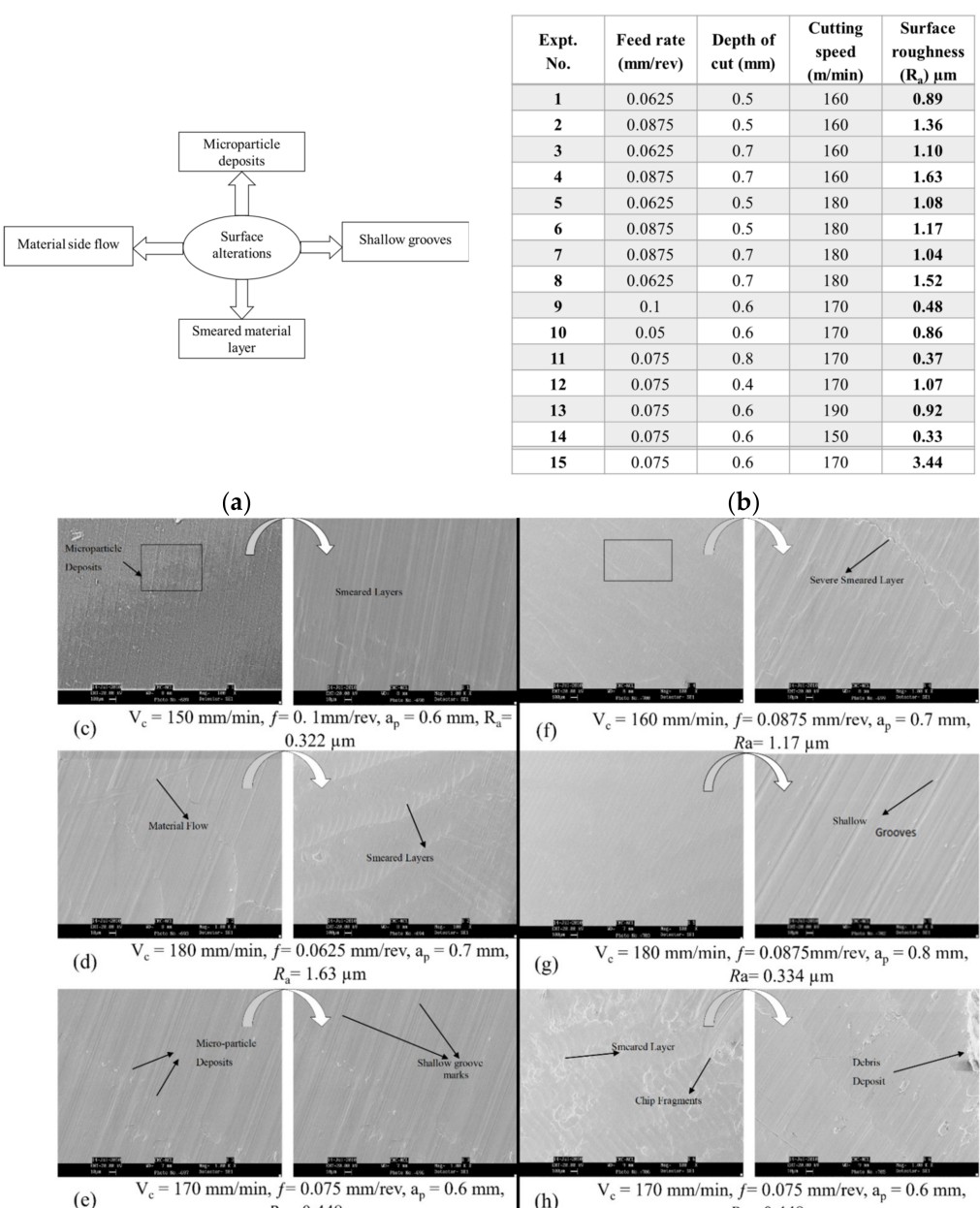

**(b)**

| Expt. No. | Feed rate (mm/rev) | Depth of cut (mm) | Cutting speed (m/min) | Surface roughness ($R_a$) μm |
|---|---|---|---|---|
| 1 | 0.0625 | 0.5 | 160 | 0.89 |
| 2 | 0.0875 | 0.5 | 160 | 1.36 |
| 3 | 0.0625 | 0.7 | 160 | 1.10 |
| 4 | 0.0875 | 0.7 | 160 | 1.63 |
| 5 | 0.0625 | 0.5 | 180 | 1.08 |
| 6 | 0.0875 | 0.5 | 180 | 1.17 |
| 7 | 0.0875 | 0.7 | 180 | 1.04 |
| 8 | 0.0625 | 0.7 | 180 | 1.52 |
| 9 | 0.1 | 0.6 | 170 | 0.48 |
| 10 | 0.05 | 0.6 | 170 | 0.86 |
| 11 | 0.075 | 0.8 | 170 | 0.37 |
| 12 | 0.075 | 0.4 | 170 | 1.07 |
| 13 | 0.075 | 0.6 | 190 | 0.92 |
| 14 | 0.075 | 0.6 | 150 | 0.33 |
| 15 | 0.075 | 0.6 | 170 | 3.44 |

**Figure 2.** (**a**) Type of surface alterations observed and (**b**) process parameters and surface roughness for machining Ti6Al4V. (**c**–**h**) Demonstration of surface alterations due to machining of titanium (Ti-6Al-4V) alloy.

The analysis of the machined surface damage performed using SEM revealed the formation of grooves on the machined surface. These grooves may result from the indentation of worn-out tool tips into the material surface. These grooves are seen throughout the surface along the feed spacing with a higher depth of cut of 0.8 mm, shown in Figure 2f,h. Machining was performed at a high cutting speed of 180 m/min. Another reason for the formation of these grooves along the feed spacing could be the abrasion of the strain-hardened material on the cutting tool.

The material displaced by the trailing edge of the cutting tool attached to the feed mark's location appears as burrs. In the case of the machined surfaces of the titanium

(Ti6Al4V) alloy, while machining at a high cutting speed, i.e., 180 m/min, such a phenomenon can be seen.

To compare the impact of the cutting speed on the surface alterations during machining, the workpiece was machined at a low cutting speed of 50 m/min and the other workpiece at 170 m/min, shown in Figure 2e. It was found in both cases that the surfaces showed microparticle deposits; however, the size of the microparticles was smaller for the surface generated at a higher cutting speed (170 m/min). Moreover, there was no substantial elevation in the surface finish, which was 0.448 μm at 170 m/min, as seen in Figure 2h, but it is still on the lower side for the machined surfaces produced at 50 m/min, as revealed in Figure 2f, i.e., 0.322 μm.

This observation of increasing the surface roughness with the increasing speed from Figure 2b follows the behavior of increasing cutting forces suggested by Soloman in high-speed machining. However, the surface roughness decreases due to the increased thermal softening effect that causes more smearing and side flow of material due to high-temperature generation. The surface roughness values increase gradually as the feed rate increases. The lowest surface roughness value obtained was 0.5 μm at the initial machining, and this increased slightly during the machining and reached up to 1.3 μm at the end of the machining. The surface finish after machining was lower than during the initial machining due to chipping wear of the nose radius and clearance face greater than the wear on the flank surface. The work material reacts differently during machining than steel at a high depth of cut and a low cutting speed. The alloying element does not allow for an increase in temperature during machining; because of the low temperature, the friction is low for the tool–workpiece interface, which gives a good surface finish. A good surface finish was observed at a low feed rate and a high depth of cut, while the surface finish deteriorated at higher feed rates and a lower depth of cut. During machining at a higher feed rate, plastic deformation increased, causing higher frictional conditions at the tool–chip interface, and thus, it led to a higher surface roughness. The assessment of the subsurface integrity involved the examination of the microhardness beneath the surface and the measurement of residual stresses on the machined surface. Initially, the subsurface integrity was assessed in terms of the degree of work hardening obtained from the microhardness examination of the machined sub-surface, and then it was statistically analysed.

It was observed from the microhardness profiles in Figure 3e–g that the pattern of microhardness variation is almost similar for all the samples until 200 μm depths beneath the surface; later, few variations are seen on the microhardness profiles beyond 200 μm to 500 μm from Figure 3a–d. It was observed from Figure 3e that the microhardness was raised considerably for a 60 μm depth in all the samples. This was then gradually decreased and reached bulk microhardness. As far as the effect of cutting speed is concerned, the microhardness values were higher (390 HV) when machining was carried out at 160 m/min at the lowest feed rate of 0.0625 mm/rev; on the other hand, the machined sample produced at a higher cutting speed of 180 m/min with same feed rate showed relatively lower values of microhardness (367 HV) beneath a 60 μm depth. The microhardness at a 20 μm depth on the cross-section of the machined subsurface was measured for different samples using various indentation loads ranging from 0.05 kg to 0.3 kg, as seen in Figure 3h. A range of indentation loads and distances between indentations were tested, and an optimized indentation distance at a lower load was selected for the Vickers hardness test [27]. A diagonal length was noted for each microhardness value corresponding to the indentation load for nine replicated samples for the improved accuracy of the microhardness values.

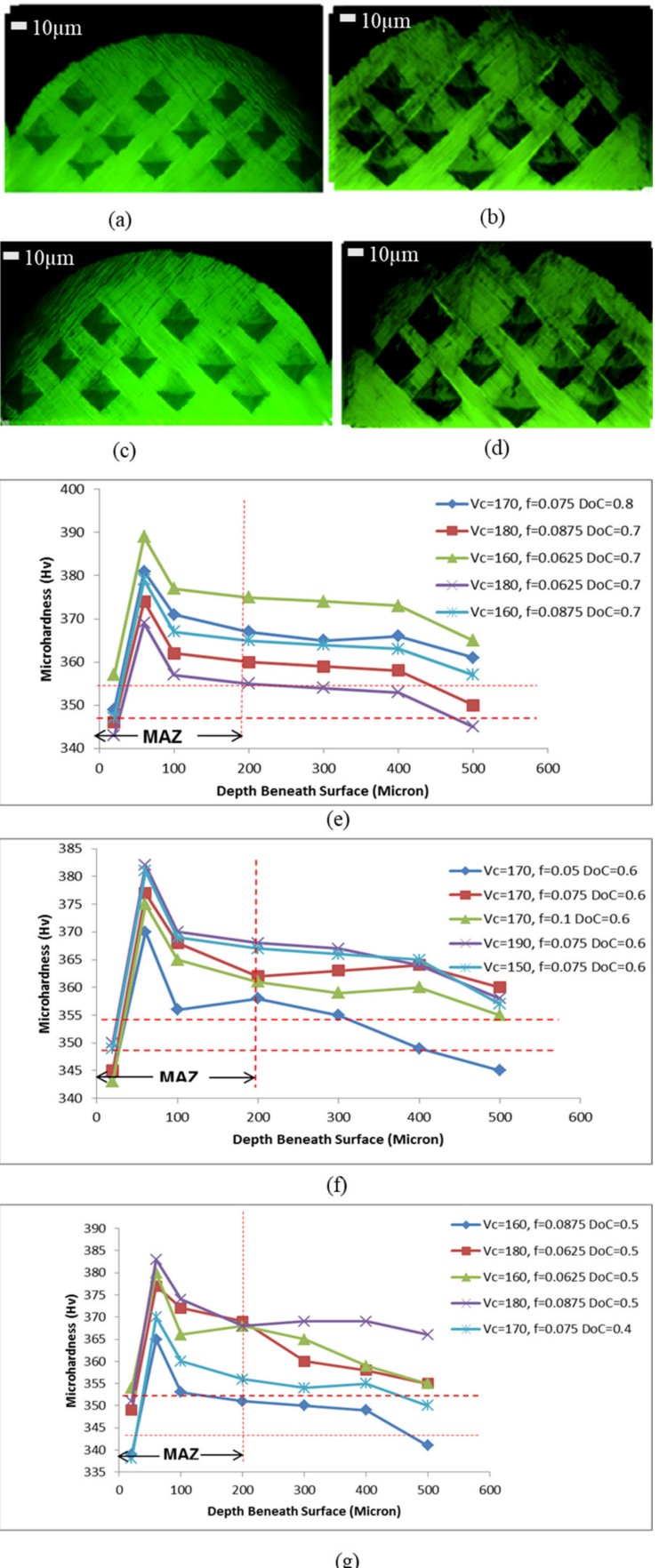

**Figure 3.** *Cont.*

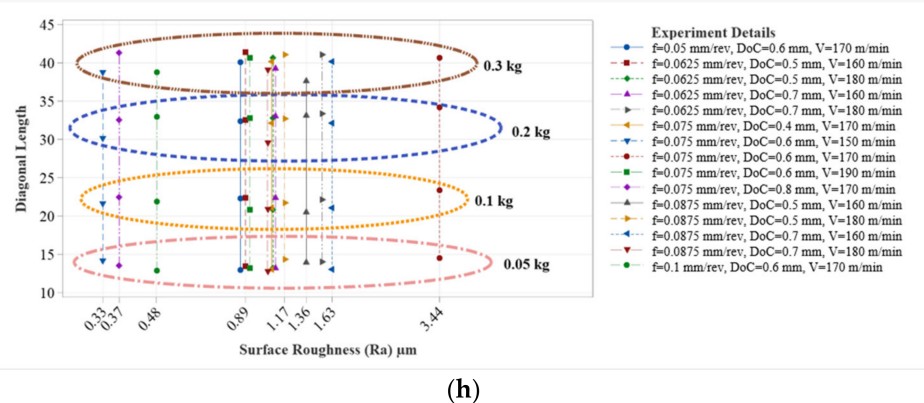

(**h**)

**Figure 3.** (**a–d**) Indentation photographs and (**e–g**) microhardness profiles of the machined specimens. (**h**) Diagonal length of indents at a 20 µm depth versus the surface roughness for different indentation loads.

This behavior of work hardening can be explained as follows: At a higher cutting speed, the temperature generation in the machining region is higher, which causes the machined surface layer to soften thermally. Therefore, at the 180 m/min speed, the machined surface shows a microhardness lower than the bulk hardness due to the prevailing thermal softening effect at 20 µm beneath the surface. However, this temperature effect is diminished as the depth increases the plastic deformation, showing more work hardening. However, as the depth beneath increases to 100 µm, the impact of thermal softening is predominant, and the microhardness approaches the bulk value. This effect can be explained by the material characteristics, i.e., the thermal conductivity of titanium alloy. The heat enters the near-surface layer (20 µm) and does not penetrate much to a 60 µm depth. Therefore, the region at 60 µm is less influenced by the thermal effect. However, the heat beyond this depth, i.e., 100 µm, does not transfer to the surroundings. The heat remains in the core due to the poor heat conduction ability of this material; hence, the thermal influence is more dominating beyond a 100 µm depth.

Figure 3f shows that the thermal softening effect dominates at 20 µm below the machined surface such as Figure 3e, and the later mechanical influence causes an increase in the microhardness at 60 µm beneath the surface. At a 20 µm depth, the feed's effect is significant and distinguished as the effect of the thermal influence is greater at 0.1 mm/rev than at 0.075 mm/rev. In this case, the microhardness was 342 HV when machining was performed at 0.1 mm/rev, increasing to 345 HV at 0.075 mm/rev.

*3.2. Discussion of the Effect of the Thermal Behavior on the Microstructure of AM of Ti6Al4V*

Table 2 summarizes the heat dissipation and the typical cooling rate in SLM, DED, and EBM, which indicates that the cooling rate is similar in all three AM processes; however, the thermal history, i.e., the build temperature, plays a vital role in determining the microstructure and hence the mechanical properties of the components manufactured from each of the processes. Figure 4 compares the cooling rate to the linear energy density. It can be observed that the SLM process has a lower linear energy density and a higher cooling rate, as well as a higher hardness and higher thermal gradient leading to columnar grains. In contrast, DED has a higher linear energy density, leading to a low cooling rate and a lower hardness; a lower temperature gradient results in equiaxed and columnar grains. In EBM, the component is continuously held at a high temperature leading to a different microstructure compared to SLM and DED, discussed in the latter part of this section. For EBM, the linear energy density is extremely high. Hence, the cooling rate is low beyond the building temperature due to which this behavior is similar to heat treatment, and the hardness is increased.

**Table 2.** Thermal behavior of different processes for Ti6Al4V.

| Thermal Properties | SLM | DED | EBM |
|---|---|---|---|
| Dominant heat dissipation | Conduction | Conduction and force convection | Conduction |
| Cooling rate (K/s) | $1.7 \times 10^4$ [28] | $7 \times 10^4$ [29] | $10^3 - 10^5$ [30] |

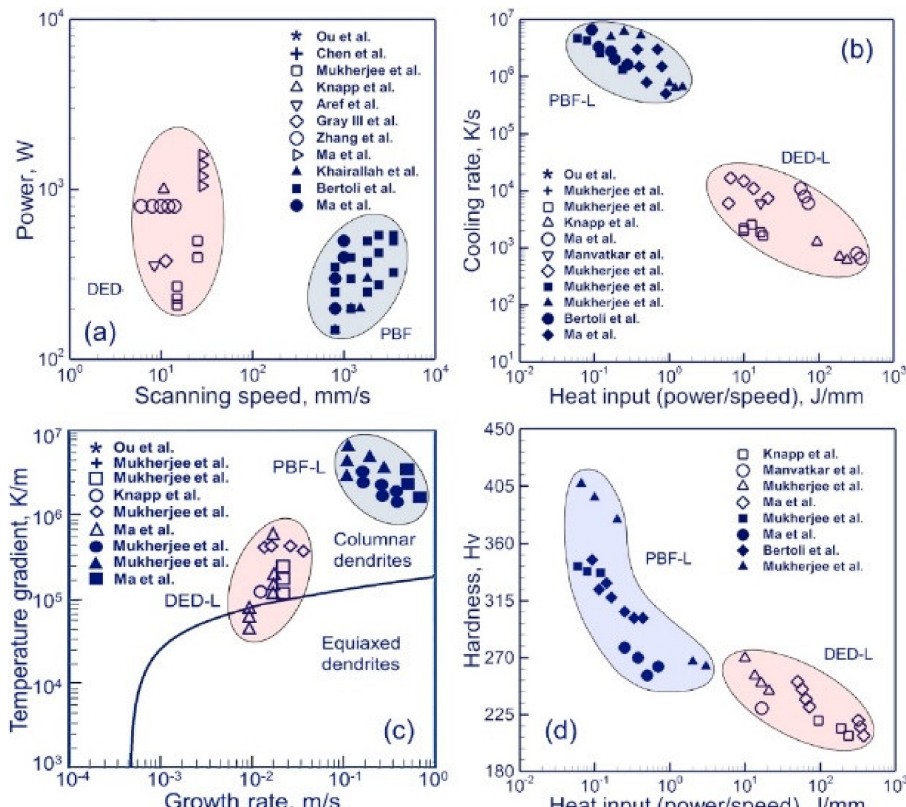

**Figure 4.** Comparison of SLM and DED: (**a**) power versus scan speed, (**b**) cooling rate versus heat input, (**c**) temperature gradient versus growth rate, and (**d**) hardness versus heat input [31–40].

### 3.2.1. Phase

For Ti6Al4V, a cooling rate above 410 °C/s forms the pure martensite phase $\alpha'$, whereas a cooling rate between 410 °C/s to 20 °C/s forms a dual phase $\alpha + \beta/\alpha'$, and a cooling rate below 20 °C/s does not form the $\alpha'$ phase at all. DED and SLM typically have the $\alpha'$ phase in excess due to a higher cooling rate of about 410 °C/s, whereas a dual phase is formed in EBM due to the high build temperature. Initially, until the temperature drops to the building temperature, EBM also produces the $\alpha'$ phase; however, due to maintaining it at the building temperature and not releasing it to room temperature, the $\alpha'$ phase starts converting to the dual phase, which occurs between 400 to 800 °C.

If the building temperature exceeds 700 °C, the complete $\alpha'$ phase will convert to a dual phase. After the component is released from its build temperature and cooled to room temperature, no phase transformation occurs. A similar case can be replicated for SLM with a low scan speed and laser power to produce a dual phase. The volume fraction of the phases and the corresponding microhardness can also be calculated with the help of a numerical model, which depends on the cooling rate.

### 3.2.2. Grains

The martensite phase is formed within the β grains, which are columnar upon solidification of the melt pool of Ti6Al4V; hence, the features of these grains directly affect the martensite phases' morphology; the scan speed decreases, and the length of the columnar

grains increases. In the DED of Ti6Al4V, long columnar grains form, except for when high power is used. The grain growth direction is perpendicular to the fusion line or the molten pool as the maximum temperature gradient is at the fusion line. Hence, a BCC lattice structure grows. The size of the grains is larger than the layer thickness as epitaxial grain growth occurs, i.e., the grains grow at the top of the previous layer due to remelting and re-solidification; hence, they exhibit a large aspect ratio with the size varying from 1 to 20 mm in length and 0.2 to 4 mm in width. The grain size consisting of the martensite and the dual phase grows with an increase in the energy density. The SLM and EBM methods lead to a slow cooling rate of the top layer due to the low thermal conductivity of the Ti6Al4V, as the heat gets trapped between the layers. However, the cooling rate for DED is high; hence, the martensite phase can be observed near the top layer. This also leads to the evolution of the microstructure, leading to physical properties such as superior elongation but a lower strength on the top layers. In EBM, grain coarsening occurs when the part is held at a temperature of 700–800 °C, and the grain size increases.

### 3.3. Defects and the Microstructure and Their Effects on Mechanical Properties and Residual Stress

Figure 5 and Figure 7 represent the melt pool in DED (laser power = 1 kW and scan speed = 0.1 m/s) and SLM (laser power = 500 W and scan speed = 3 m/s), showing that the melt pool in DED is approximately 3 mm long, while the melt pool in SLM is only 0.3 mm in length. The temperature at which the Ti6Al4V completely solidifies is 1693 K. The shape of the SLM melt pool is flat because of the thin layers, and it is 10% of the DED, which is hemispherical and dependent on the surface tension and volume of the melted drops. A more significant thermal gradient means considerable surface tension, which results in a large pool size and eventually a higher residual stress generated in the component. Hence, the SLM of Ti6Al4V has the lowest residual stress as compared to DED and EBM, while EBM, due to high energy, has a larger melt pool and higher residual stresses, which can be verified from Figures 6 and 7a,b. The higher residual stress directly affects the fatigue strength, and the fatigue crack increases, as seen in Figure 5, when the residual stress is more elevated. Heat treatment and preheating of the fusion bed reduce the residual stress.

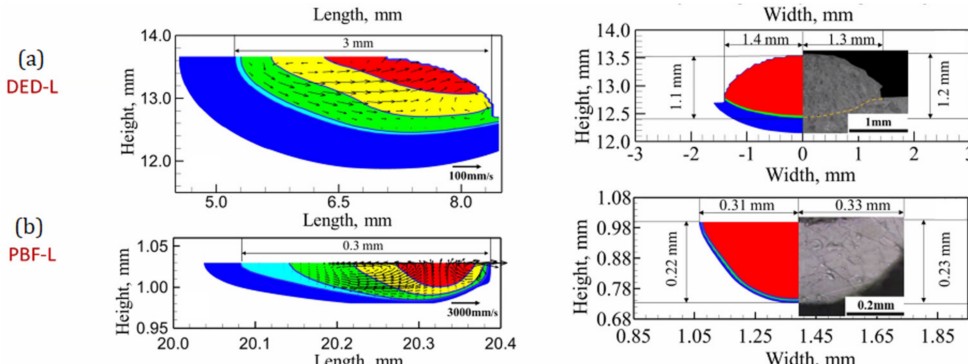

**Figure 5.** Comparison of the melt pool in (**a**) DED and (**b**) SLM to learn about the thermal history and cooling rate [41].

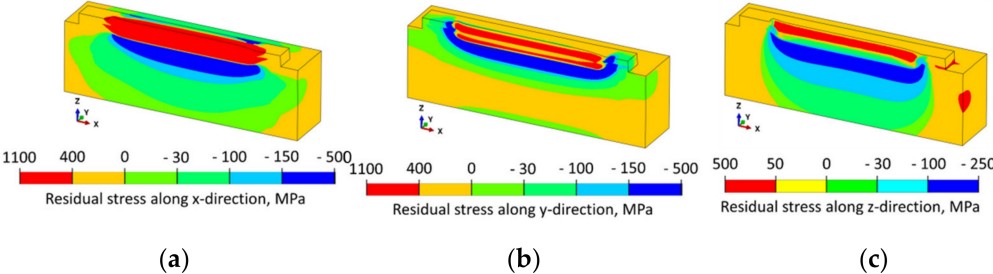

**Figure 6.** Residual stress distribution for Ti6Al4V along the (**a**) x, (**b**) y, and (**c**) z directions of the two-layer deposit [42].

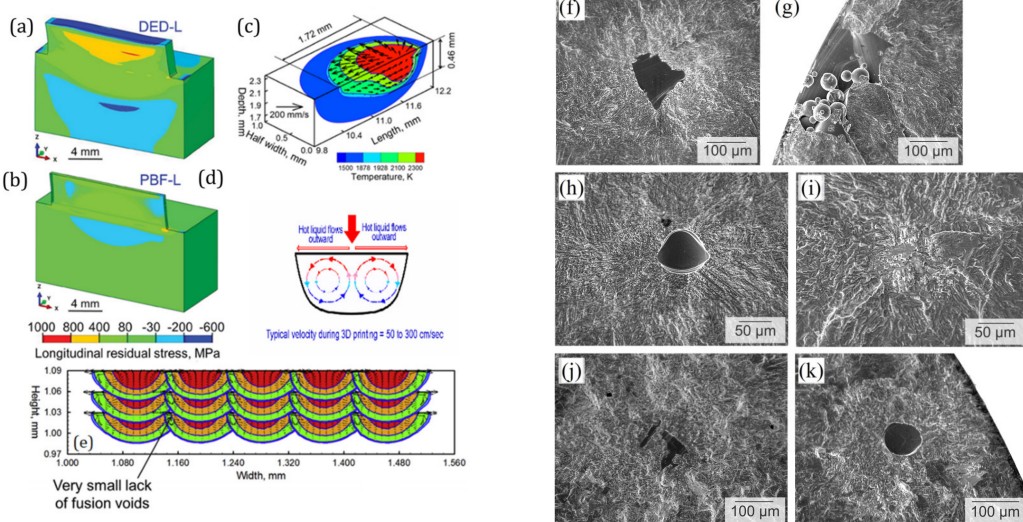

**Figure 7.** Longitudinal residual stress distribution in (**a**) DED and (**b**) SLM. (**c**) Temperature and velocity distributions during the deposition of the 2nd layer of Ti64. (**d**) Motion of the molten pool. (**e**) Transverse-sectional view of the molten pools for three layers and five hatch builds of Ti6Al4 V using 60W laser power 1000 mm/s [41,43]. (**f**) Lack of fusion defects in SLM with a stress amplitude of 353 MPa and $1.4 \times 10^7$ cycles to failure. (**g**) Lack of fusion in SLM caused by inadequate consolidation of the powder particles with unmolten particles visible at the defect's inner surface with a stress amplitude of 342 MPa and $1.56 \times 10^5$ cycles to failure. (**h**) Approximately circular pore created by trapped gas during SLM with a stress amplitude of 342 MPa and $5.59 \times 10^7$ cycles to failure. (**i**) $\alpha$-phase defect type observed in SLM with a stress amplitude of 560 MPa and $1.17 \times 10^7$ cycles to failure. (**j**) Lack of fusion defects observed in EBM with a stress amplitude of 250 MPa and $2.3 \times 10^8$ cycles to failure. (**k**) Approximate circular pore in EBM with a stress amplitude of 250 MPa and $1.6 \times 10^8$ cycles to failure [44].

### 3.3.1. Porosity

Uncontrolled porosity leads to reduced ductility and sites for the nucleation of microcracks. In DED (0.1%), spherical porosity occurs due to feeding gases entrapped between the layers. In SLM (nonoptimized—0.23%, optimized—0.08%) and EBM (0.17%), it occurs due to excessively high energy leading to the pores settling at the bottom of the layer, also called keyhole porosity. For printing Ti6Al4V with a lesser porosity using SLM and EBM, the optimal parameters with high energy must be employed as a higher energy would fully melt the powder. Hence, the viscosity would be reduced, filling in the pores. However, the lack of fusion pores, as seen in Figure 7e, is irregular in shape with sharp tips and is more detrimental. It can be partially avoided by increasing the linear energy density, which means a higher laser power and a low scan speed, as seen in Figure 7c, which in turn generates an optimal melt pool to avoid the lack of fusion and also to resist the residual stresses caused in the Ti6Al4V part. Heat treatment does not reduce porosity; instead, HIP, dense metallic powder, and a high linear energy are effective techniques to reduce the percentage of pores in the part; however, the final geometry of the region will be altered.

### 3.3.2. Surface Roughness

Surface roughness is a vital defect to be minimized to increase the endurance life of the manufactured component. It can be seen from Figure 8 that the surface texture of the EBM-produced part is superior compared to SLM; however, the surface roughness of EBM is inferior to the SLM-produced piece because of loose powder being bonded with the surface in EBM due to the slow cooling produced due to the high thermal energy. DED produces the best surface quality compared to SLM and EBM as there is a smaller possibility of partial melting under optimal conditions. To improve the surface finish in SLM and

EBM, fine powder must be used; in DED, an optimized hatch spacing, a low feed rate, and a high scan speed must be used. Ultimately, post-machining is the ideal choice to reduce surface roughness.

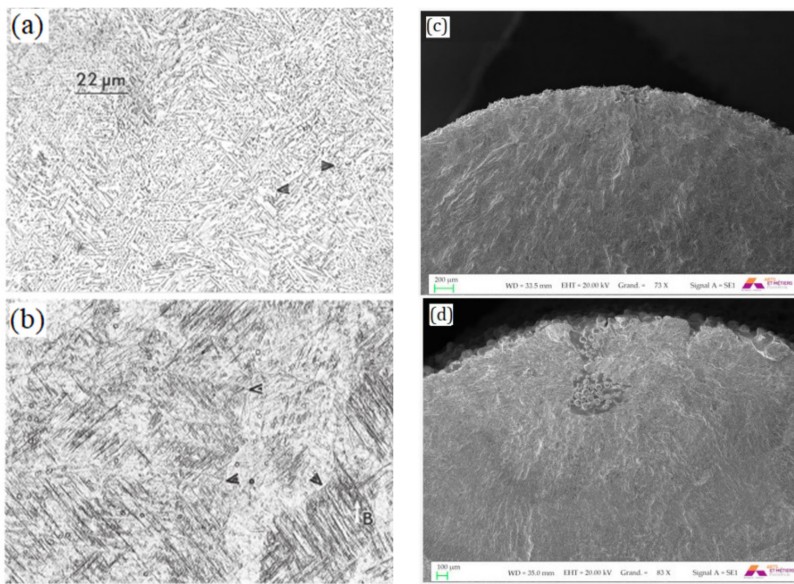

**Figure 8.** Comparison of the microstructures for (**a**) EBM and (**b**) SLM and the surface roughness for (**c**) SLM and (**d**) EBM for Ti6Al4V [45].

### 3.3.3. Tensile

As temperature increases, the resistance to plastic flow decreases; hence, the elongation percentage drops at higher temperatures. If strength increases, ductility decreases, as seen in Figure 9. For machined parts produced by AM methods, the elongation percentage in EBM is higher, and the ultimate tensile strength and yield strength are lower than SLM and DED, as seen from Figure 9c,d. SLM and DED showcase the formation of the $\alpha'$ nonequilibrium martensitic phase, which produces lattice strains that increase strength and exhibit a high dislocation density, resulting in hardening. Due to heat treatment, grain coarsening improves the elongation percentage and reduces strength. HIP increases the ductility as the microstructure becomes compact. SLM and DED pose the $\alpha'$ martensite phase, which exhibits the highest strength and low plasticity. At the same time, the dual phase exists in EBM-produced parts, which consist of a dendritic microstructure, mainly columnar and equiaxed, where the former shows a higher strength. Titanium is a reactive metal that needs to be manufactured in a controlled atmosphere as oxidation reduces its ductility. As oxidation increases, surface hardness, UTS, and yield strength increase due to dislocation pinning, and the color changes from silver to straw to blue. During the recycling and handling of the powder, the powder is generally sieved, which exposes it to an uncontrolled atmosphere and leads to the formation of oxides. Hence, recycled powder must be avoided. The anisotropic behavior of the tensile property can be correlated to the printing direction and microstructure. For horizontally printed parts, the tensile load acts along the short axis of the β grains, making it easy to fail. In contrast, the tensile load works along the longer axis for a longitudinally printed part, unveiling higher elongation.

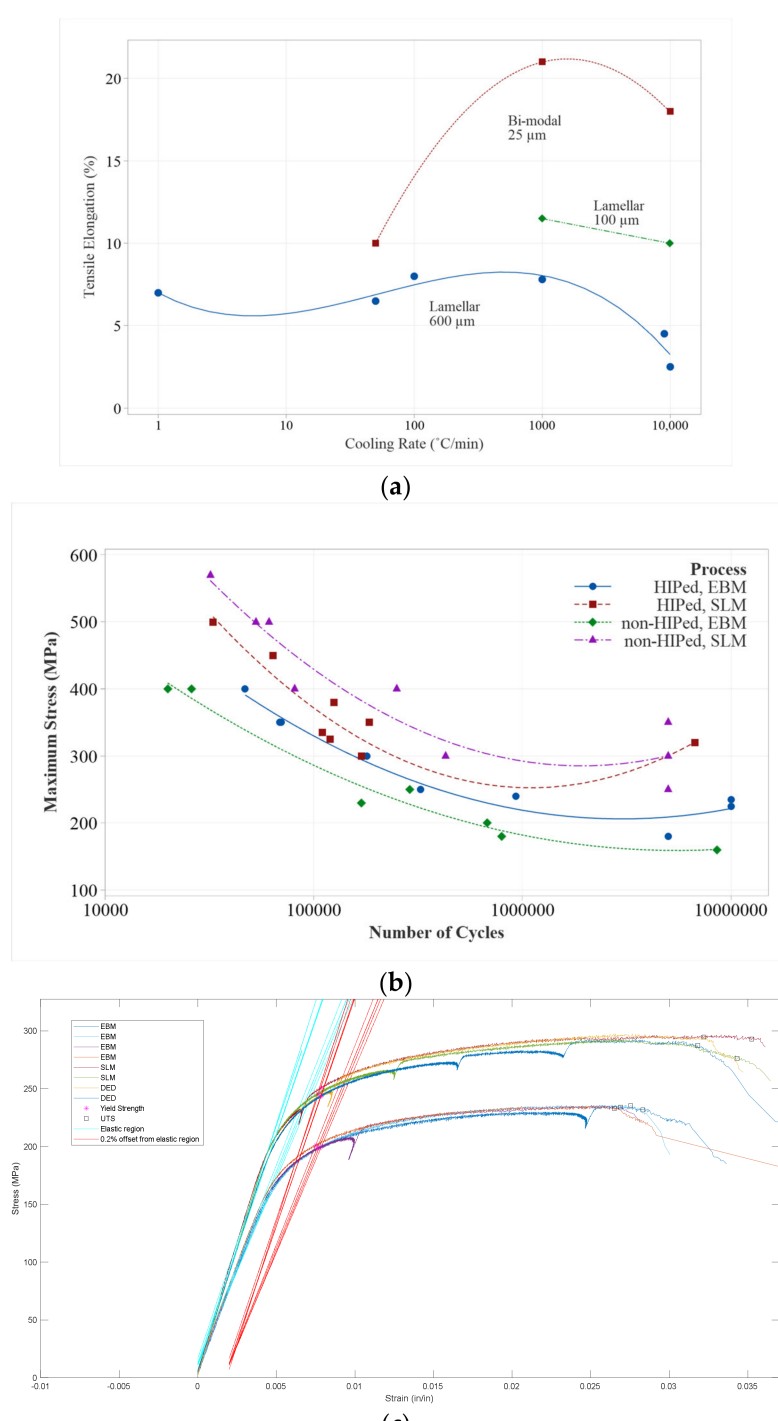

(**a**)

(**b**)

(**c**)

**Figure 9.** *Cont.*

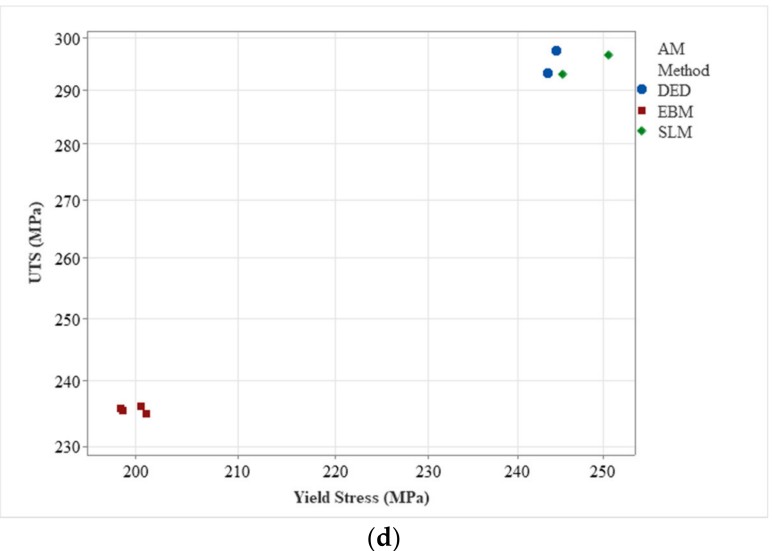

(**d**)

**Figure 9.** (**a**) Dependence of tensile elongation on the cooling rate of Ti6Al4V for different grains. (**b**) Fatigue life of HIPed and non-HIPed SLM and EBM samples without notches (stress concentration factor ($K_t$) =1). (**c**) Stress–strain curve for the dog bone coupon manufactured using SLM, DED, and EBM. (**d**) Comparison of yield stress and UTS for SLM manufactured using SLM, DED, and EBM.

### 3.3.4. Fatigue

For as-built parts prepared by SLM, the fatigue strength is high, as shown in Figure 9b, because of the presence of the $\alpha'$ martensite phase, but they have a low fatigue toughness due to plastic strain as compared to EBM. EBM has a higher fatigue toughness due to the $\alpha + \beta$ dual phase [45]. HIP improves the fatigue strength and toughness as it reduces the porosity and makes the part dense, reducing the chances of cracking. Hence, the surface roughness, which behaves similar to short cracks, and porosity, which is majorly governed by shear, have a significant impact on the fatigue properties of the part as they serve as the initiation points for the development of cracks in the component. The defect's shape is also essential in determining the fatigue strength. The imperfections depend on the type of defect, such as the lack of fusion sharp tips, porosity, sphericalness, etc., and stress concentration occurs in the lack of fusion defect leading to a low fatigue strength. Since residual stress only occurs in SLM and DED, residual stress in tension is not desirable; however, residual stress in compression enhances the fatigue properties.

### 4. Conclusions

This paper comprehensively explains the behavior of Ti6Al4V during conventional high-speed machining and various additive manufacturing (AM) processes, which are essential to determine the behavior of the physical properties and the microstructure peculiarities in AM as compared to conventionally manufactured components.

Various alterations/defects, such as shallow grooves, microparticle deposits, white layers, etc., are formed during the machining of Ti6Al4V. It was observed that the feed rate and the depth of cut are the main parameters that influence the occurrence and magnitude of these defects on the machined surfaces of Ti6Al4V alloy.

At high cutting speeds between 150 m/min and 190 m/min, an increase in the shear stresses in the machining zone leads to plastic deformation on the top layer of the machined surface. Thus, white layers are seen on the machined surface. Microparticle deposits are visible at a higher cutting speed of 170 m/min at moderate feed rates. Shallow grooves were seen throughout the surface along the feed spacing with a superior depth of cut of 0.8 mm when machining was performed at an elevated cutting speed of 180 m/min.

By identifying the gap between high-speed dry machining experiments and closely observing the graphical plots of the surface finish results with various cutting parameters, meticulous considerations were made while deciding on the combinations of the cutting parameters to obtain the best possible surface finish. The variety of surface properties in machining and AM of the current experiments and previous studies working on Ti6Al4V alloy have been studied closely. The optimized combination was zoomed-in on in this paper to identify the crux of behavioral changes in the part's surface finish. Thus, this micro-level study presented the reasons for the observed changes in the surface finish, defects, and hardness and suggestions for improving them.

**Author Contributions:** Conceptualization, S.P.Y. and R.S.P.; methodology, S.P.Y. and R.S.P.; software, S.P.Y.; validation, S.P.Y. and R.S.P.; formal analysis, S.P.Y. and R.S.P.; investigation, S.P.Y. and R.S.P.; resources, R.S.P.; data curation, S.P.Y. and R.S.P.; writing—original draft preparation, S.P.Y.; writing—review and editing, S.P.Y. and R.S.P.; visualization, S.P.Y.; supervision, R.S.P.; project administration, R.S.P.; funding acquisition, R.S.P. All authors have read and agreed to the published version of the manuscript.

**Funding:** This research received no external funding.

**Data Availability Statement:** Not applicable.

**Conflicts of Interest:** The authors declare no conflict of interest.

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
