# Peer review of "Manufacturing Methods Induced Property Variations in Ti6Al4V Using High-Speed Machining and Additive Manufacturing (AM)"

_metals, doi:10.3390/met13020287_

Round 1

Reviewer 1 Report

The manuscript entitled Manufacturing Methods Induced Property Variations in 2 Ti6Al4V using High-Speed Machining and Additive Manufacturing (AM) reports the detailed and exhaustive examination of the behavior of titanium alloy Ti6Al4V under the common high speed machining in comparison of that under AM conditions.

The paper is well-structured and written in very good English language. The main message of the article is clear. I think the manuscript can be published provided some corrections will be made.

Thus, there is a strange numbering of paragraphs (lines 124 and 127) which should be corrected.

Also, the captions to Figures 4 and the manuscript text are mixed, which makes it difficult to understand the text. Figures with the captions and the text should be separated.

The heading of Section 3.2 should be changed; it does not correspond to the content of the Section.

The numbering of the Section title on the line 438 should also be corrected.

Author Response

We agree with the reviewer’s comment and thank you for making such a critical and close observation of the submitted manuscript We have incorporated the changes and modified the manuscript according to the reviewers’ suggestions. These suggestions by the reviewers have been taken seriously by us, and to eliminate such errors, we have revised the entire manuscript, which has definitely raised the quality of the manuscript.

P1: There is a strange numbering of paragraphs (lines 124 and 127) which should be corrected.

Answer: The necessary changes have been made

P2: Also, the captions to Figures 4 and the manuscript text are mixed, which makes it difficult to understand the text. Figures with the captions and the text should be separated.

Answer: The authors have incorporated the suggested changes by the reviewer in the manuscript. 

P3: The heading of Section 3.2 should be changed; it does not correspond to the content of the Section.

Answer: This was a typographical error and the necessary changes have been made. 

The numbering of the Section title on the line 438 should also be corrected.

Answer: This was a typographical error in the numbering, and we have made the necessary changes. 

Reviewer 2 Report

In this manuscript, the author describes the experimental investigations in high-speed dry machining and additive manufacturing (AM) of titanium alloy. It was observed that the Ultimate tensile strength and Yield Strength were approximately 20% more in SLM and Direct Energy Deposition as compared to Electron Beam Melting .However,the current paper needs to be revised before it is considered for publication. The main comments are as follows:

1.Too much content in the introduction section suggesting the author to refine.

2. There is a problem with the title on the fourth page asking the author to make corrections.

3.“Error! Reference source not found.. There is a problem with the formatting of this sentence, please ask the author to correct it.

4. Please enlarge the text in Figure 4 (e), (f), (g) for the reader's convenience.

5. Please mark (c) and (d) in the left part of Figure 8.

6. The conclusion section has too much content suggesting the author to refine.

Author Response

The authors would like to thank the esteemed reviewer for precisely pointing out the shortcomings of the paper and therefore helping to improve the quality of the manuscript. The authors have taken all the comments raised by the esteemed reviewer into due consideration and have made the necessary changes in the manuscript. 

P1.Too much content in the introduction section suggesting the author to refine it.

Answer: The introduction has been shortened and refined, and the outline of the paper has been stated in the introduction.

P2. There is a problem with the title on the fourth page, asking the author to make corrections.

Answer: We have corrected the typographical error, incorporating the suggested changes. 

P3. “Error! Reference source not found..” There is a problem with the formatting of this sentence, please ask the author to correct it.

Answer: This comment was not clear.

P4. Please enlarge the text in Figure 4 (e), (f), (g) for the reader's convenience.

Answer: The authors have incorporated the suggested changes by the reviewer, and the text in the figure has now been enlarged. 

P5. Please mark (c) and (d) in the left part of Figure 8.

Answer: The authors have now marked the sub-captions in the figure as suggested by the reviewer.

P6. The conclusion section has too much content suggesting the author to refine.

Answer: The conclusions have been shortened and refined. 

Reviewer 3 Report

The study is interesting and the reviewer believes that the study can be published in this journal.

However, there are some points that need to be improved:

- Various indentation loads where used but the range described in the text is different of Figure 4 h). Should be '0.05 kg to 0.3 kg'.

- In each indentation load for high diagonal length of indents there is a abrupt increase of surface roughness. How do you explain this behavior?

- In Figure 10 b) missing a legend identifying the curves of yield stress and elongation.

- How was evaluated the fatigue strength of material regarding the Figure 10. The authors need to perform fatigue tests for different conditions of fatigue tests (e.g. R ratio and level of loads) to be possible to assess the fatigue strength of materials as a function of AM processes.

Author Response

We are extremely thankful to the esteemed reviewer for pointing out the errors in the manuscript, which the authors have now rectified as per the suggestions. 

P1: Various indentation loads where used but the range described in the text is different of Figure 4 h). Should be '0.05 kg to 0.3 kg'.

Answer: The typo of  '0.05 kg to 0.5 kg' in the text has been corrected to '0.05 kg to 0.3 kg'.

P2: In each indentation load for high diagonal length of indents there is a abrupt increase of surface roughness. How do you explain this behavior?

Answer: We would like to thank the reviewer for bringing this up, as this particular graph was misleading. We have now modified the plot to enhance the clarity. The point to be conveyed was not to establish a relationship between the microhardness and the surface roughness, but the motive behind the graph was to point out whether the surface roughness of the sample has any impact on the microhardness, and if it has then at what indentation load does it impact the most. In the earlier graph, the information was misleading, however, in the new graph, we separated and systematically presented the data and eliminated any ambiguity. 

P3: In Figure 10 b) missing a legend identifying the curves of yield stress and elongation.

P4: How was evaluated the fatigue strength of material regarding the Figure 10. The authors need to perform fatigue tests for different conditions of fatigue tests (e.g. R ratio and level of loads) to be possible to assess the fatigue strength of materials as a function of AM processes.

Answer for P3. and P4. - The older figure Figure 10 b) was in the manuscript by mistake (dummy image), which has now been replaced by our experimental plot. We apologize for the typographical error. We have now presented the results of the fatigue tests performed for SLM and EBM HIP'ed and as-printed samples. 

Round 2

Reviewer 3 Report

The authors corrected the manuscript taking into account the reviewers comments.